# Comparative Study of Gut Microbiota in Wild and Captive Giant Pandas (*Ailuropoda melanoleuca*)

**DOI:** 10.3390/genes10100827

**Published:** 2019-10-20

**Authors:** Wei Guo, Sudhanshu Mishra, Chengdong Wang, Hemin Zhang, Ruihong Ning, Fanli Kong, Bo Zeng, Jiangchao Zhao, Ying Li

**Affiliations:** 1Farm Animal Genetic Resources Exploration and Innovation Key Laboratory of Sichuan Province, Sichuan Agricultural University, Chengdu 611130, Sichuan, China; guochina2005@126.com (W.G.); mishra.sudhanshu30@gmail.com (S.M.); about2006@163.com (R.N.); kfl229@163.com (F.K.); apollobovey@163.com (B.Z.); 2School of Laboratory Medicine/Sichuan Provincial EngineeringLaboratory for Prevention and Control Technology of Veterinary Drug Residue in Animal-origin Food, Chengdu Medical College, Chengdu 610500, China; 3China Conservation and Research Center for the Giant Panda, Ya’an 611830, Sichuan, China; wang_2019pandas@sina.com (C.W.); zhang_2019pandas@sina.com (H.Z.); 4Department of Animal Science, Division of Agriculture, University of Arkansas, Fayetteville, AR 72701, USA

**Keywords:** captivity, gut microbiome, giant pandas, enzyme activity, antibiotic resistance genes, metagenomics, diversity, metal tolerance genes, virulence factors

## Abstract

Captive breeding has been used as an effective approach to protecting endangered animals but its effect on the gut microbiome and the conservation status of these species is largely unknown. The giant panda is a flagship species for the conservation of wildlife. With integrated efforts including captive breeding, this species has been recently upgraded from “endangered” to “vulnerable” (IUCN 2016). Since a large proportion (21.8%) of their global population is still captive, it is critical to understand how captivity changes the gut microbiome of these pandas and how such alterations to the microbiome might affect their future fitness and potential impact on the ecosystem after release into the wild. Here, we use 16S rRNA (ribosomal RNA) marker gene sequencing and shotgun metagenomics sequencing to demonstrate that the fecal microbiomes differ substantially between wild and captive giant pandas. Fecal microbiome diversity was significantly lower in captive pandas, as was the diversity of functional genes. Additionally, captive pandas have reduced functional potential for cellulose degradation but enriched metabolic pathways for starch metabolism, indicating that they may not adapt to a wild diet after being released into the wild since a major component of their diet in the wild will be bamboo. Most significantly, we observed a significantly higher level of amylase activity but a lower level of cellulase activity in captive giant panda feces than those of wild giant pandas, shown by an in vitro experimental assay. Furthermore, antibiotic resistance genes and virulence factors, as well as heavy metal tolerance genes were enriched in the microbiomes of captive pandas, which raises a great concern of spreading these genes to other wild animals and ecosystems when they are released into a wild environment. Our results clearly show that captivity has altered the giant panda microbiome, which could have unintended negative consequences on their adaptability and the ecosystem during the reintroduction of giant pandas into the wild.

## 1. Introduction

The giant panda (*Ailuropoda melanoleuca*) has been listed as an endangered animal species for decades (The IUCN Red List of Threatened Species, 2016) but concerted, integrated efforts to protect the species has led to a 31.7% population increase and a recent upgrade in conservation status from “endangered” to “vulnerable” (The IUCN Red List of Threatened Species, 2016). Captive breeding has been one of the main integrated approaches used towards protecting the giant panda and increasing their population size. There are 21.8% of giant pandas living in zoos worldwide (http://www.chinadaily.com.cn/china/2017-11/18/content_34698154.htm; http://www.iucnredlist.org/details/712/0). The gut microbiome of giant pandas has been implicated in their health and disease status [1]. For example, gastrointestinal diseases have been reported to be a primary cause of death in giant pandas [2,3]. In addition, it has been speculated that giant pandas depend on gut microbiota to obtain nutrition from the highly fibrous bamboo [4,5]. Habitat is shown to be an important factor affecting the gut microbiomes of animals [6,7]. Previous studies have demonstrated that captivity is associated with altered gut and skin microbiomes in several different species spanning a wide range from mammals to fish [8,9,10,11,12,13,14,15,16,17,18,19]. Therefore, we hypothesized that captivity alters the gut microbiome of giant pandas. Recently the Chinese government initiated a program to reintroduce giant pandas to the wild. Given the important roles that a gut microbiome plays in mammalian health and disease [20], it is critical to characterize and compare the gut microbiome between captive and wild giant pandas, particularly to evaluate if captive breeding altered the gut microbiomes of giant pandas and whether these changes affect the fitness of these animals after release to the wild. Previous studies have suggested that gut microbial compositions are different between captive and wild giant panda populations [1,5,21,22,23,24]. However, subject to sequencing technique, sample size or bioinformatics analysis methods, it was difficult to obtain effective and accurate information on the differences between the gut microbiotas of wild and captive giant pandas in the existing literature. Especially, there is little information available on the potential effect of captive breeding on gut microbiotas function of giant pandas. For example, whether captivity alters the efficiency of cellulose degradation in captive giant pandas. To better understand the effects of captivity on the gut microbiome of giant pandas, we used 16S rRNA marker gene and whole-genome shotgun sequencing to characterize and compare the gut microbiomes of captive and wild giant pandas.

## 2. Methods

### 2.1. Sample Collection, DNA Extraction, and Sequencing

In this study, fecal samples were collected from both wild and captive giant pandas. Fecal samples were collected from wild giant pandas at the Fengtongzhai National Nature Reserve (FNNR, n = 14) and the Wolong National Nature Reserve (WNNR, n = 67) by experienced trackers, immediately frozen in liquid nitrogen, and stored at −80 °C until later use. Fecal samples from captive giant pandas were collected from the China Conservation and Research Center for the Giant Panda (CCRC; n = 49) and also stored at −80 °C until later use. Demographics for all pandas sampled in this study are shown in Appendix A. Seven tetra-microsatellites including GPL-60, gpz-47, gpz-20, GPL-44, GPL-29, GPL-53, and gpz-6 were used to distinguish the wild individuals, and this DNA analysis was performed by Qiao et al., therefore the samples of wild giant pandas were from their research (Appendix A) [25].

DNA was extracted from the fecal samples using the Mo Bio PowerFecal DNA isolation kit (Mo Bio Laboratories, Carlsbad, CA, USA) according to the manufacturer’s instructions. Variable region 4 (V4) forward primer: GTGYCAGCMGCCGCGGTAA, reverse primer: GGACTACHVGGGTWTCTAAT. The PCR reaction (50 µl total volume) contained 2 µL DNA (20 ng), 19 µL PCR grade water, 25 µL 2× Es Taq Master Mix (CW BIO, Beijing, China) (including 2× Es Taq Polymerase, 0.075 µM Mg^2+^, and 10 µM dNTP mix), 2 µL (0.4 µM) forward primer, and 2 µL (0.4 µM) reverse primer. PCR was performed at 94 °C for 1 min, followed by 30 cycles of 94 °C for 20 s, 59 °C for 25 s, and 68 °C for 45 s, followed by a final extension at 68 °C for 10 min. Variable region 4 was sequenced at the Beijing Genomics Institute (Beijing, China).

Shotgun metagenomic sequencing was performed at Novogene (Beijing, China). DNA libraries were constructed according to Illumina’s instructions. Briefly, DNA was sheared to 300–400 bp fragments, and the DNA from each individual sample was barcoded uniquely. Since most of the demographic data for the wild pandas are unknown, we intended to choose a wider range of samples from the captive samples to see if the environment is still the major driver of the gut metagenome. Therefore, we randomly chose three cubs (≤2-year-olds), one sub-adult (>2-year-olds and <5-year-olds) and three adults (≥5-year-olds and ≤20-year-olds) (including three males and four females) for metagenome analysis (Appendix A). Correspondingly, seven random samples were collected from wild giant pandas at the Fengtongzhai National Nature Reserve and were chosen for comparative analysis of metagenomics with captive giant pandas. Fourteen samples were sequenced on an Illumina HiSeq platform. Sequencing depth, quality control, and pre-processing details are listed in Appendix A.

The datasets used in this study are accessible from the National Centre for Biotechnology Information Sequence Read Archive (SRA; http://www.ncbi.nlm.nih.gov/sra) [26], accession bio project numbers: PRJNA356809 and PRJNA358755.

### 2.2. S Amplicon Data Analysis

16S rDNA amplicon raw sequences were processed by using mothur v1.39 following the MiSeq SOPs [27]. Briefly, high-quality sequences with a length distribution from 250 to 500 bp, without ambiguous bases, and homo-polymers shorter than 8 bp were retained. Then, the clean reads were analyzed using QIIME v1.9.0 [28]. These sequences were then clustered into operational taxonomic units (OTUs) against the Greengenes v13.5 reference OTU (97% similarity) using a closed reference OTU-picking approach. Chimeras were identified and removed using UCHIME [29], and, to further reduce sequencing error, chloroplast sequences and singleton OTUs were also removed. Before computing α and β diversity metrics, we rarefied the OTU table to 8000 sequences per sample (the minimum number of sequences observed in a single sample). We then computed the Shannon index and observed species α diversity metrics to assess α diversity and Jaccard and Bray–Curtis distances to assess β diversity. ‘RandomForest’ v.4.6-7 [30] was used to estimate the key variables for differentiating the microbiomes of the wild giant pandas from those of captive pandas. Linear discriminant analysis coupled with effect size (LEfSe) (LDA score > 2.5 and *p*-value < 0.05) was used to identify OTUs differentially represented between wild and captive giant pandas at the genus level [31]. To assess which factors affect the gut microbiomes of wild giant pandas, we performed PERMONOVA analysis based 16S dataset to examine whether different collecting time (2015/2013), individuals affect, and population (Wild (FNNR)/Wild (WNNR)) were significantly associated with gut microbial communities of giant pandas.

### 2.3. Metagenome Analysis

#### 2.3.1. Quality Control

Reads were first de-multiplexed using unique per-sample barcodes. Illumina adaptors were trimmed using cutadapt 1.3 [32]. Burrows–Wheeler Aligner (BWA) [33], Samtools [34] and Bam2fastq (https://gsl.hudsonalpha.org/information/software/bam2fastq) were used to remove reads that mapped to Moso Bamboo [35] (i.e., the diet) and the Giant Panda [4] (i.e., the host) genomes using default parameters. Reads were trimmed at the first low quality (Q < 20) base in a sliding window using Trimmomatic [36]. Reads shorter than 75 bp after trimming were discarded, and only paired-end reads were used in downstream analyses.

#### 2.3.2. De Novo Assembly and Construction of a Non-Redundant Gut Microbiome Gene Set

IDBA-UD was used to assemble the high-quality reads into contigs with a large k-mer range (20–120) [37]. The k-mer value that resulted in the longest N50 value was chosen for the remaining scaffolds. The per-sample assembled contigs were then used to predict ORFs with MetaGeneMark (prokaryotic GeneMark.hmm version 2.8) [38]. CD-HIT was used to define the non-redundant “panda gut microbiome gene set” with a 95% identity cutoff [39].

#### 2.3.3. Read Normalization

Before aligning the sequences against the non-redundant panda gut microbiome gene set, we reduced bias caused by uneven sequence depth by randomly sub-sampling all samples to the same number of read (9, 237, 293 paired-end reads) using seqtk-master (https://github.com/lh3/seqtk).

#### 2.3.4. Gene Abundance Profiling

To align paired-end reads against the non-redundant panda gut microbiome gene set, we used SOAPaligner v2.21 (http://soap.genomics.org.cn/soapaligner.html) with default parameters (‘-2 -m 400 –x 600 –n 5 –r 1 –v 2 –M 4′). Reads that were mapped at multiple positions were discarded. We calculated the unique per-sample reads mapping to a single gene as, while the length of the gene was defined as. Then, for gene ‘g,’ gene abundance (*Ab*(*g*)) was calculated as follows:

If the relative abundance in each sample is *Rb*(*g*), the computation is:$$Ab(g)=\frac{R(g)}{L(g)}$$
$$Rb(g)=\frac{R(g)}{L(g)}\cdot \frac{1}{\sum_{i=1}^{n}\frac{Ri}{Li}}$$

#### 2.3.5. Gene Functional Annotation

To annotate the metagenomes, we aligned the non-redundant panda gut microbiome gene set against the Kyoto Encyclopedia of Genes and Genomes (KEGG) online database using BLAST with the parameters “for prokaryotes in [a] representative set” and “single-directional best hit” [40]. We used Pfam hidden Markov models HMMER3 [41] to identify the Carbohydrate-Active ENZYmes hits with E-values <1 × 10^−3^ (E-values <1 × 10^−5^ for protein sequences <80 aa) and a coverage >0.3.

#### 2.3.6. Analysis of Resistance Genes and Virulence Factors

Using BLASTP (v 2.4.0+), the non-redundant panda gut microbiome gene set was further searched against the Antibiotic Resistance Genes Database (http://ardb.cbcb.umd.edu/documentations.shtml#bdb) [42], Aclame Database (http://aclame.ulb.ac.be/) [43], the BacMet database containing antibacterial biocide- and metal-resistance genes (version 1.1; http://bacmet.bio medicine.gu.se/) [44], and the virulence factor database (full dataset B; http://www.mgc.ac.cn/VFs/main.htm) [45]. Hits with an e-value <1 × 10^−5^, percent identity ≥80%, and alignment length ≥50 bp were considered positive.

#### 2.3.7. α Diversity, β Diversity, and Statistical Comparisons of the Abundances of CAZy Families, Select Gene Groups, and Viruses

Using QIIME v1.8.0 [28], we determined the number of unique functional genes present in each sample after rarefying to the lowest number of mapped reads, and, as an additional proxy for α diversity, computed the Chao1 index on the rarefied data [46]. Jaccard, Euclidean, and Bray–Curtis distances were calculated on observation matrix tables containing abundance information on KOs, CAZy families, and antibiotic resistance genes (ARGs), and we first normalized all KO, CAZy families, and ARGs in each sample to sum to 1. We then used the PAST v.3.1 data analysis package [47] to build principle coordinate analysis (PCoA) plots. We also created a UPGMA-clustering tree of CAZyme using the relative abundance of CAZy families. and compared the relative abundances of these families involved in cellulose (GH5, GH6, GH7, GH9, GH44, GH45, GH48) degradation [5,48] in wild and captive giant pandas. We also compared the relative abundance of genes encoding CAZymes related to starch degradation (GH13, GH14, GH15, GH57, GH97, GH119) (http://www.cazy.org/) [49]. Finally, we compared the relative abundances of antibiotic resistance genes (ARGs), mobile genetic elements (MGE’s), metal resistance genes (MRGs), and virulence factors in wild and captive giant pandas.

All figures were generated in R using the packages “boxplot,” “barplot,” “pheatmap” and “plot.” Mann–Whitney U tests of significance using GraphPad Prism 5 software (GraphPad Software, Inc., USA) were used to determine significant differences between groups.

#### 2.3.8. Phylogenetic Analysis of Metagenome Dataset

We analyzed the metagenome sequences with the MG-RAST pipelines [50]. Sequences were classified by using the Refseq database with the following parameters: Maximum e-value of 1 × 10^−5^, the minimum percent identity of 60, and minimum alignment length of 30. The relative abundance of the taxonomic profiles was calculated based on the genus-level classification and used to calculate the Bray–Curtis distances using the PAST v.3.1 data analysis package [47].

##### Enzyme Activity Assay

To determine the cellulose and starch degradation activity of gut microbiota in giant pandas, cellulase and amylase activity of fecal samples from captive and wild giant pandas was quantitatively estimated by calculating degradation capacity (i.e., producing 1 µg reducing sugars per gram fecal sample per minute) [51,52]. Here, we selected fecal samples of shotgun metagenomics sequencing to measure enzyme activity (Appendix A). The cellulase and amylase activity was determined using cellulase anthrone colorimetry [53] and α-amylase (3,5-dinitrosalicylic acid method [54] activity kit (Comin Biotechnology, Suzhou, China), respectively.

##### Ethics Approval and Consent to Participate 

This study was approved by the Institutional Animal Care and Use Committee of the Sichuan Agricultural University under permit number DKY-B20130302. All authors read and approved the submission of this manuscript. 

## 3. Results

### 3.1. Captivity Reduced Microbial Diversity and Altered the Gut Microbiome Structure of Giant Pandas

We characterized the fecal microbiomes from 49 healthy captive pandas raised at the CCRC for the Giant Panda (Ya’an and DuJiangyan, Sichuan Province, China) and from 81 wild giant pandas living in the FNNR (Baoxing, Sichuan Province, China) and the WNNR (Wenchuan, Sichuan Province, China) (Appendix A). Comparing the fecal microbiomes of captive and wild pandas, we observed significant differences in both α and β diversity. The Shannon diversity index and the number of observed operational taxonomic units (OTUs) of gut microbiotas in captive pandas were both significantly lower than that in wild pandas (Figure 1A,B and Appendix A). Figure 1C shows the top 50 most discriminatory OTUs identified by a random forest classifier, which was able to predict wild pandas from the captive pandas with 98% accuracy. At the phylum level, the wild panda microbiome was dominated by Proteobacteria, followed by Bacteroidetes and Firmicutes. In contrast, the guts of captive pandas were dominated by Firmicutes, followed by Proteobacteria (Appendix A). In wild pandas, Pseudomonas was the most abundant genus detected, while Streptococcus and Enterobacteriaceae were the most abundant genera in captive pandas (Appendix A). LEFSe analysis identified 45 and 38 enriched bacterial taxa in the wild and captive giant pandas, respectively (Appendix A). The LDA score of Pseudomonas, Streptococcus, and Enterobacteriaceae were the highest among all genus identified by LEFSe.

Principal coordinates analysis (PCoA) based on the Bray–Curtis (Figure 1D and Appendix A) and Jaccard (Figure 1E and Appendix A) distances revealed distinct clustering patterns between wild and captive pandas (Mann–Whitney test, *p* < 0.01).

Of note, it is challenging to assign the animal origins of these 81 wild samples, i.e., it is likely that some samples were collected from the same giant panda. A total of 27 individual wild giant pandas were identified according to DNA analysis (Appendix A). We randomly selected a sample as the representative of each speculated giant panda to re-analyze the data. Consistently, we observed reduced community diversity and shifted community structure in the captive giant pandas (Appendix A). In addition, we also performed a phylogenetic analysis of the metagenome data using MG-RAST, and the result also showed distinct gut microbiomes between captive and wild giant pandas (Appendix A). The PERMANOVA analysis showed that collection time did not affect the gut microbiome of the wild giant pandas (F = 1.46, *p* = 0.13), but individuals did, with the genotypes (F = 1.674 *p* = 0.037) as the most influential (Appendix A).

### 3.2. Captivity Reduced Functional Potential for Cellulose Degradation but Increased Metabolic Pathways for Starch Metabolism

To examine the difference in the functional potential of wild and captive panda microbiomes, we selected 14 samples (seven from each group) for shotgun metagenomic sequencing. De novo assembly of 310.5 million reads with IDBA-UD [37] resulted in an average of 119,645 contigs and an N50 length of 1262 bp. Gene predictions with MetaGeneMark [38] yielded 2.1 million genes, and 1.1 million non-redundant genes were verified with CD-HIT using a criterion of 95% identity [39]. These genes were queried against the Kyoto Encyclopedia of genes and genomes (KEGG) database using KAAS (KEGG Automatic Annotation Server [40]). The diversity of functional genes was significantly lower in captive pandas than in wild pandas (Appendix A), and the number of reads mapping to genes with unknown functions was higher in wild pandas (Appendix A). Not surprisingly, distinct clusters were also observed in PCoA visualizations of Euclidean (Figure 2A), Jaccard (Appendix A), and Bray–Curtis (Appendix A) distances. Specifically, the gut microbiome of wild pandas is enriched in genes from pathways involved in amino acid, cofactor, vitamin, and lipid metabolism, cellular processes, and xenobiotics biodegradation and metabolism. Conversely, the gut microbiome of captive pandas is enriched in genes from nucleotide metabolism pathways (Figure 2B). Notably, genes involved in starch and cellulose degradation were differentially abundant in wild and captive pandas, with wild panda microbiomes containing a significantly higher abundance of putative endoglucanase (EC:3.2.1.4; Appendix A) and captive panda microbiomes containing a significantly higher abundance of putative α-amylase (EC:3.2.1.1; Appendix A).

Deeper statistical analysis based on assessment of carbohydrate-active enzymes (CAZymes), especially genes involved in cellulose and starch degradation, revealed that the microbiomes of captive pandas encode significantly fewer CAZy family enzymes involved in degrading cellulose and hemicellulose (Figure 2C) but encode significantly more CAZy family enzymes involved in starch degradation (Figure 2D), corroborating our previous observations (i.e., the enrichment of endoglucanase (EC:3.2.1.4; Appendix A) and α-amylase (EC:3.2.1.1; Appendix A) in the gut microbiome of giant pandas. Furthermore, UPGMA clustering of CAZymes families yielded complete separation of wild and captive pandas (Figure 2E).

### 3.3. Captivity Enriched Antibiotic Resistance Genes in the Microbiomes of Captive Pandas

Because antibiotic resistance and the widespread incorporation of antibiotic resistance genes (ARGs) into pathogens via mobile genetic elements (MGEs) pose a severe threat to public health [55], we hypothesized that ARGs might be similarly significant in the context of captivity, and determined whether ARGs are increased in the gut microbiomes of captive pandas. Querying the Antibiotic Resistance Genes Database [42] revealed that although present in the microbiomes of both wild and captive pandas, there were significantly more ARGs (both in terms of type and pure numbers) in captive pandas (Figure 3A,B). For example, ARGs with activity against Aminoglycosides/Glycylcyclines/Macrolides/β Lactams, Puromycin, Bacitracin, Doxorubicin/Erythromycin, Deoxycholate/Fosfomycin, and Cephalosporin were significantly enriched in captive pandas (Figure 3C). Euclidean-, Bray–Curtis-, and Jaccard-based PCoA plots constructed on ARGs revealed distinct clusters separating wild and captive pandas (Appendix A). Similarly, a search against Aclame Database [43] demonstrated that the type and the abundance of MGEs were also significantly higher in captive pandas (Figure 3D,E).

### 3.4. Captivity Enriched Heavy Metal Tolerance Genes in the Microbiomes of Captive Pandas

Metals have long been used as antimicrobial agents and may exert co-selection pressure for ARGs [56]. Therefore, we queried genes in our shotgun dataset against the BacMet database [44] to identify metal resistance genes (MRGs). We identified genes involved in resistance to 18 metals, including copper, nickel, cadmium, cobalt, and zinc. Consistent with our observations on ARG presence and abundance, metal resistance genes (with the exception of chromium, bismuth, and aluminum) were more abundant in the gut microbiome of captive pandas (Figure 4A,B).

### 3.5. Captivity Enriched Virulence Factors in the Microbiomes of Captive Pandas

In addition to differences in ARGs, we observed significant differences in virulence factors present in the gut microbiomes of wild and captive pandas. By aligning the shotgun data against the virulence factor database (VFDB, full dataset B) [45], we determined that the overall abundance of virulent genes was significantly higher in captive pandas (Figure 4C). Genes encoding flagella, peritrichous flagella, the type III secretion system (TTSS), and O-antigen were especially enriched in captive pandas (Figure 4D). 

### 3.6. Enzyme Activity Between Giant Pandas

The metagenomics-based results indicated that the potential capability of wild giant pandas in digesting cellulose is significantly higher than captive giant pandas. In addition, captive panda microbiomes have a significantly higher potential capability of digesting starch. To further verify this finding, we evaluated and compared the cellulase and amylase activity between captive and wild giant pandas. The activity of the cellulase of wild giant panda feces was significantly higher than captive giant panda feces (Mann–Whitney test, *p* < 0.05, Figure 5A). In contrast, we observed a significantly higher level of amylase activity (Mann–Whitney test, *p* < 0.05, Figure 5B) in captive giant panda feces than those in wild giant pandas.

## 4. Discussion

Recently, a similar study has revealed that the diversity and gut microbial communities of the wild and translocated giant pandas were significantly different from the captive pandas [23], and our findings also support it. However, Yao et al. primary focused on the variation of the gut microbiome in giant pandas after release into the wild environment. Therefore, their study did not describe the differences between the gut microbiotas of the captive and wild giant panda populations in detail. Especially, there is no comparative analysis of metagenomics and enzyme activity for the gut microbiome of captive and wild giant panda populations. Here, we demonstrate that the diversity, composition, functional potential, enzyme activity, and presence of ARGs, MGEs, MRGs, and virulence factors in the fecal microbiomes of wild and captive giant pandas differ significantly. Diets are different between captive and wild giant pandas. Giant pandas are fed steamed grain mixture, animal products, and bamboo, etc. in zoos [57]. However, the wild giant pandas are exclusive bamboo specialists with almost 99% of its diet being bamboo [58]. In addition, giant pandas can feed on more kinds of bamboo in the wild. Considering diet and phylogeny as the main factors in shaping the gut microbiota of animals [59,60,61], the decrease in cellulose-degrading pathways and enrichment for starch metabolism in the captive giant pandas are likely due to the consumption of easily digestible food (e.g., steamed corn and wheat bread) in addition to bamboo, as suggested by a previous study (i.e., pandas preferred some bamboo species or parts of bamboo, which contained more protein and less cellulose and lignin) [62]. Our data is consistent with recent reports that the activity of cellulase is low, but amylase activity is high in captive giant panda (Figure 5) [5], and cellulase are significantly lower compared to those in other herbivores [22,63]. Whether decreased diversity in giant pandas is linked to disease as it is in humans [64] is unknown, and future studies should aim to fill this knowledge gap. Several studies have reported altered microbiomes in a variety of animal species as well as increased incidence of certain diseases in captivity [13,65,66]. What remains to be determined, however, is whether or not the two are connected. Nevertheless, this possibility, together with the fact that the loss of certain gut microbes may not be recovered over several generations [67], is worrisome.

Yao.et al‘s study indicated that the diversity and gut microbial communities of translocated giant pandas become closer to wild giant pandas after releasing into the wild, and suggested candidate pandas for reintroduction should live in the translocation site for an additional year prior to release [23]. Our results based on metagenomics and in vitro experimental assay demonstrate that the capability of captive giant pandas in digesting cellulose is significantly lower than that of wild giant pandas. However, captive panda microbiomes have a significantly higher capability of digesting starch. Considering the different dietary composition between captive (more starch intake) and wild giant pandas may significantly affect the composition of gut microbiotas, we suggest that candidate pandas should be provided a similar diet and living environment with the wild giant panda populations for a long time prior to release. This could help giant pandas adapt to the wild environment and diet as soon as possible after release into the wild.

The enrichment of ARGs, MGEs, MRGs, and virulence factors genes in the captive pandas could be caused by several different routes. Animals in captivity receive regular veterinary care and are continually monitored, increasing the likelihood of more frequent medications. Antibiotic administration could provide selective pressure for the acquisition of ARGs in the captive panda microbiome and maybe a contributing factor to the increased incidence of these genes noted in our study. However, analysis of the medical history of these captive giant pandas (Appendix A), revealed that antibiotics administration may not be the major source of ARGs in these pandas (Figure 3C). The shift from a remote, wild, less-polluted environment to the captive environment (Conservation and Research Center for the Giant Panda, Ya’an and Dujiangyan) polluted by anthropogenic activities is likely the major contributing factor in the enrichment of these genes. Air pollution has been shown to be an important reservoir and mediator for the spread of ARGs [68] and is also an important route for heavy metal deposition [69]. Recently, air pollution in the Sichuan basin has increased remarkably, and seven cities were ranked in the top 100 polluted cities in China in 2016 (data provided by the Chinese atmospheric website, http://www.chndaqi.com/news/251991.html). Indeed, a recent study reported significantly higher levels of heavy metals such as cadmium and lead (Pb) in the feces of captive giant pandas than in the feces of wild pandas [70]), which could serve as selection pressure for MRGs in the fecal microbiomes of captive pandas. In addition to air pollution, contaminated water is likely another route of ARG spread. Bacitracin, Aminoglycoside, and β-lactam resistance genes (ARGs) are most dominant in the drinking water of China [71]. These ARGs are also present in captive giant pandas with a higher abundance (Figure 3C). ARGs can also be transferred between humans and animals [72]. Therefore, human contact with captive giant pandas during the captive breeding program could also result in the enriched ARGs in captive giant pandas. Surprisingly, most ARGs present in captive giant pandas, although with much higher abundance, are also detected in wild giant pandas. Although less polluted, we speculate that the wild habitat serves as the main source for the low ARGs in the wild giant pandas [68].

The ID and demographic information of the wild giant pandas are unknown. It is possible that some of the samples were collected from the same animal. To overcome this challenge, we assigned the 81 feces to 27 individual wild giant pandas based on unique genotypes. Consistent results were observed based on the new analysis. During our metagenomics analysis, we intended to select samples from a wider range of giant pandas (e.g., different age, gender) to minimize the potential bias caused by the unknown demographics of the wild giant pandas. To determine if different collecting time and individuals affected our data analysis we performed PERMONOVA and found that collecting time (F = 1.47, *p* = 0.136) did not affect the gut microbiome of the wild giant pandas, but individuals (F = 1.674, *p* = 0.037) did (Appendix A). It suggests that the method of selecting representative samples based on a DNA analysis is reliable.

Nevertheless, we found that captivity altered the gut microbiome of giant pandas in ways that may decrease fitness in the wild, by decreasing microbial diversity and functional potential for cellulose degradation and increasing antibiotic resistance genes, virulence factors, as well as heavy metal tolerance genes in the microbiomes of captive pandas. The spread of ARGs from animals to humans is a well-recognized zoonotic problem of concern [73]. In addition, previous studies have disclosed that captivity is associated with shaping different gut microbiomes in the wild and captive mammals, such as bharals [15], deer mice [18], Andean bears [10], primates [9], rhinos [17], Tibetan wild ass [16], sika deer [11], horses [12], and red pandas [8], etc. Obviously, captivity alters the gut microbiome composition and potential functions of mammals, and significantly reduces microbial diversity. Animals living in a zoo undergo a range of changes from wild animals. Such as reduced diet type, reduced diet diversity, reduced contact with other species, reduced home range, reduced interactions with different types of habitat types, human contact, antibiotics, and other veterinary intervention. These factors may be the main reasons for the significant difference in the gut microbiome between captive and wild animals [74].

As the major goal of captive breeding is to release captive pandas into the wild, the increased presence of such genes in the microbiomes poses another concern: The spread of the ARGs genes into the microbiomes of wild pandas and other animals could have unintended negative consequences on the ecosystem as a whole. Our data suggest that care should be taken during the human contact with giant pandas in the captive management programs to avoid the potential spread of ARGs from humans to pandas. The results we report here also suggest that appropriate and successful animal husbandry will need to take into account the effects of captivity on the gut microbiome. Captivity has been the most effective approach for endangered species conservation and is partially responsible for the upgrade in the conservation status of the giant pandas from “endangered” to “vulnerable” (The IUCN Red List of Threatened Species, 2016). However, the significantly altered microbiomes we identified in captive pandas compared to wild pandas indicate that captivity might have a profound effect on the long-term evolution and ecology of the gut microbiome in giant pandas.

## 5. Conclusions

We found that captive breeding significantly reduced gut microbial diversity, metabolic pathways, and enzyme activity for cellulose degradation, but increased antibiotic resistance genes, heavy metal resistance genes, and virulent factors. We emphasize that strategies (e.g., prebiotics, probiotics, fecal microbiota transplant) should be developed immediately to maintain and/or restore the gut microbiome diversity and metabolic potential of captive giant pandas, which may increase the fitness of these pandas for future introduction to the wild. We also argue that integrated efforts from policymakers and biologists should be taken to reduce pollution and the human-animal-environmental spread of ARGs, MRGs and virulence factor genes for the benefit of the long-term conservation of giant pandas, and for the maintenance of a robust ecosystem.

## Figures and Tables

**Figure 1 genes-10-00827-f001:**
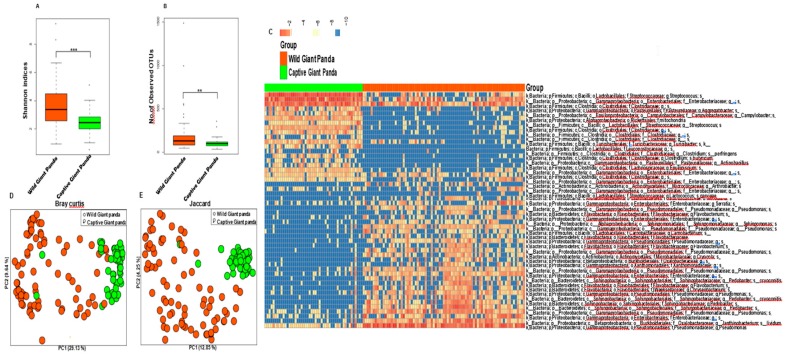
Comparison of α and β diversity of the gut microbiome of wild and captive giant pandas. The Shannon diversity index (**A**) and the number of observed OTUs (**B**) are significantly higher in wild than in captive giant pandas. Boxplot indicates the interquartile range (IQR), lines inside boxes denote the median, and error bars represent the lowest and highest values, respectively. (**C**) Heat map showing the top 50 OTUs identified by random forest as key features differentiating the microbiomes of wild and captive giant pandas. Principal coordinates analysis plots of Bray–Curtis (**D**) and Jaccard (**E**) distances demonstrate distinct gut microbiome structures and memberships between wild and captive pandas. ** *p* < 0.01, *** *p* < 0.001, Mann–Whitney U test.

**Figure 2 genes-10-00827-f002:**
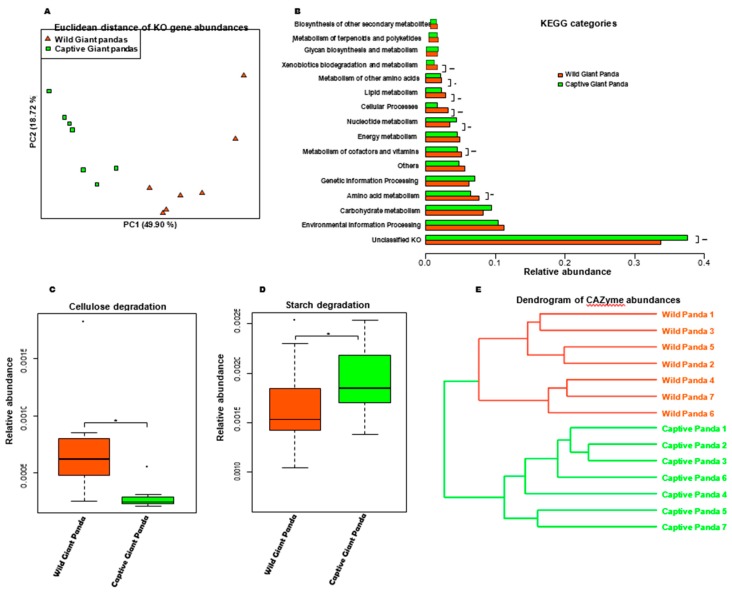
Metagenomic analysis of the functional potential of the giant panda microbiome. (**A**) Principal coordinates analysis based on Euclidean distances of all functional genes and pathways shows distinct clustering patterns between wild and captive pandas. (**B**) The relative abundances of KEGG Orthology. The relative abundance of CAZy families for cellulose (**C**) and starch (**D**) degradation. Boxplot indicates the interquartile range (IQR), lines inside boxes denote the median, and error bars represent the lowest and highest values, respectively. (**E**) UPGMA-clustered CAZyme dendrogram. * *p* < 0.05, ** *p* < 0.01, and *** *p* < 0.001, Mann–Whitney U test.

**Figure 3 genes-10-00827-f003:**
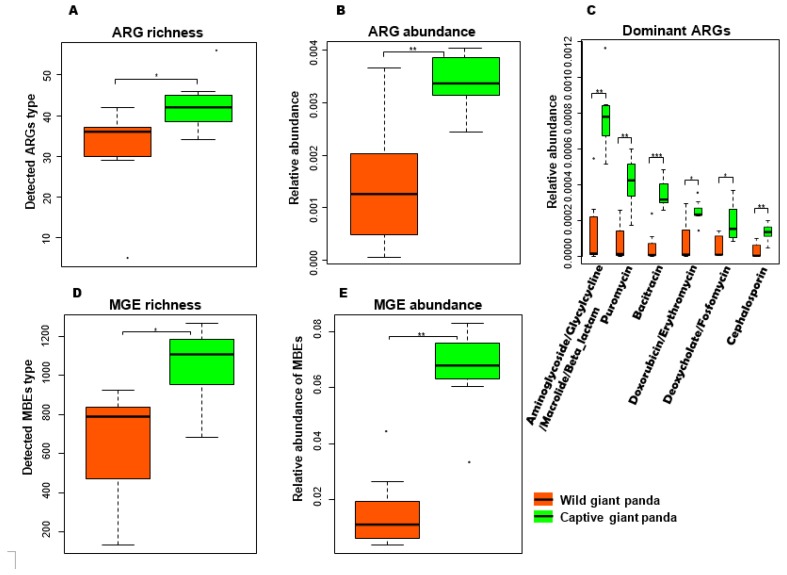
Diversity and abundance of antibiotic resistance genes (ARGs) and mobile genetic elements (MGEs) in the giant panda microbiome. (**A**) ARG diversity. (**B**) ARG relative abundances. (**C**) Comparison of ARG relative abundances in the microbiomes of wild and captive giant pandas. (**D**) MGE diversity. (**E**) MGE relative abundances. All boxplots indicate the interquartile range (IQR), lines inside boxes denote the median, and error bars represent the lowest and highest values, respectively. * *p* < 0.05, ** *p* < 0.01, and *** *p* < 0.001, Mann–Whitney U test.

**Figure 4 genes-10-00827-f004:**
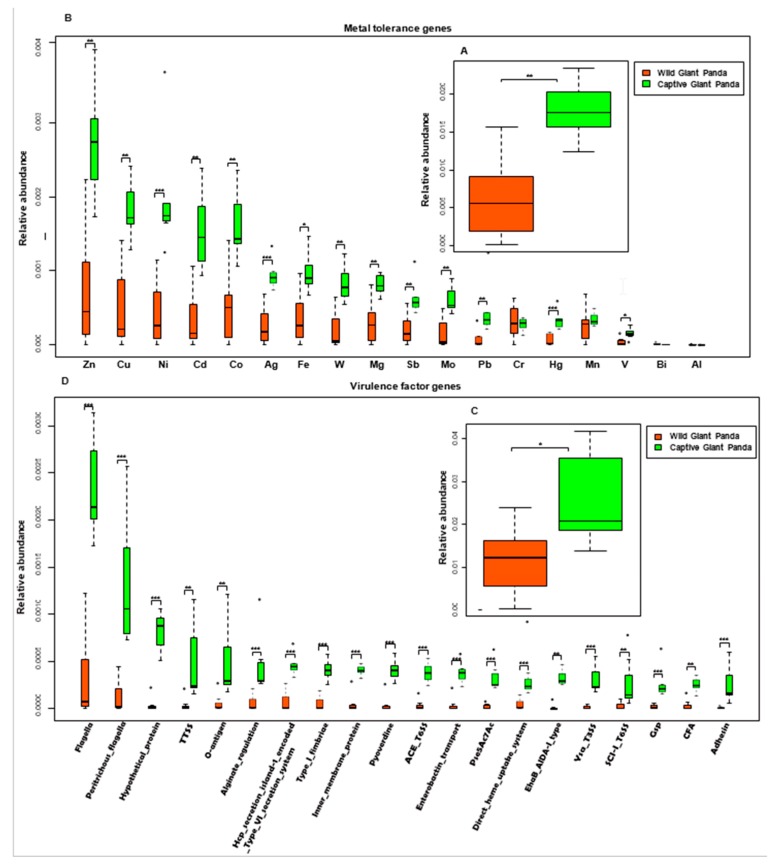
The relative abundances of metal tolerance and virulence factor genes in the giant panda microbiome. (**A**) Metal tolerance gene relative abundances. (**B**) Comparison of metal resistance gene relative abundances in the microbiomes of wild and captive giant pandas. (**C)** Virulence factor gene relative abundances. (**D**) Comparison of virulence factor gene relative abundances in the microbiomes of wild and captive giant pandas. All boxplots indicate the interquartile range (IQR), lines inside boxes denote the median, and error bars represent the lowest and highest values, respectively. * *p* < 0.05, ** *p* < 0.01 and, *** *p* < 0.001, Mann–Whitney U test.

**Figure 5 genes-10-00827-f005:**
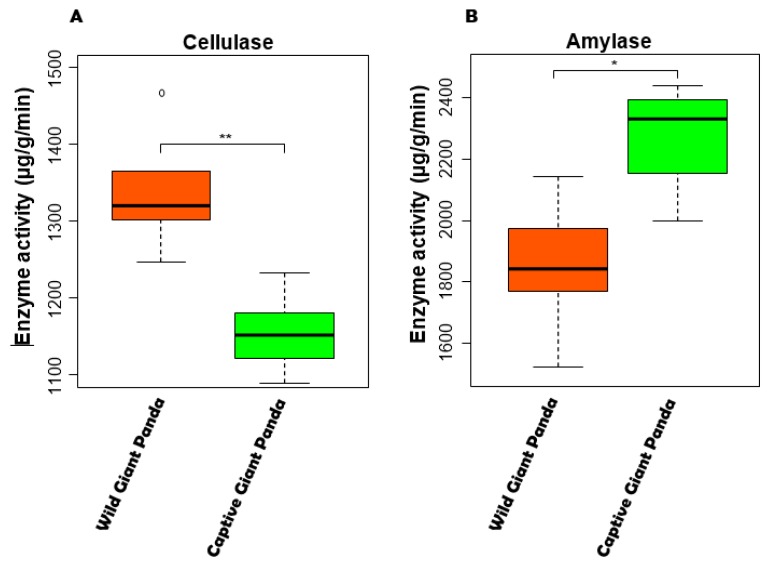
The cellulase (**A**) and amylase (**B**) activity based on the giant panda feces. * *p* < 0.05 and ** *p* < 0.01, Mann–Whitney U test.

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
