# Peer review of "Comparative Study of Gut Microbiota in Wild and Captive Giant Pandas (Ailuropoda melanoleuca)"

_genes, 2019, doi:10.3390/genes10100827_

Round 1
Reviewer 1 Report
This article explores the genetics and some metabolic aspects of captive and wild giant pandas. Overall the manuscript if of high quality, some general and specific comments follow:
Very little reference is made to similar studies in other wildlife. the authors need to address how their findings relate to knowledge and findings in microbiome studies in other species.
References need to be updated, e.g. in line 53: these are already 5 years old
Specific comments:
line 42: 21.8 suggests high precision, not "about"
line 63: english: ACCURATE
LINE 65: poor grammar
line 67: what guidance is provided? this is an overly ambitious statement.
lines 80-83: las muestras son tomadas del suelo, no saben de qué animal vienen (línea 78)
line 98: randomly
lines 205-206 correct grammar "Here samples used for shotgun..."
line 333 enzyme activity DIFFERENCES? between wild pandas
Author Response
Very little reference is made to similar studies in other wildlife. the authors need to address how their findings relate to knowledge and findings in microbiome studies in other species.
Response:We appreciate the reviewer’s encouraging comments and suggestion. We have supplemented reference that is made to similar studies in other wildlife, and discussed how our findings relate to knowledge of microbiome studies in other species( See line 445-456)
References need to be updated, e.g. in line 53: these are already 5 years old
Response:Suggestion accepted, we have updated these references(See line 59-61)
Specific comments:
line 42: 21.8 suggests high precision, not "about"
Response:Suggestion accepted, we have deleted “about”(See line 47)
line 63: english: ACCURATE
Response:Suggestion accepted, ‘accuracy’ have been replaced with ‘accurate’(See line 73)
LINE 65: poor grammar
Response:Suggestion accepted, we have corrected the poor grammar(See line 76-78)
line 67: what guidance is provided? this is an overly ambitious statement.
Response:Good point, our previous statement may be inaccurate. Here, we revised this sentence to make it more realistic(See line 78-79).
lines 80-83: las muestras son tomadas del suelo, no saben de qué animal vienen (línea 78)
Response:Fecal samples of wild giant pandas were collected by experienced conservationists of giant pandas. In addition, the faeces of giant pandas is mainly composed of undigested bamboo fragments, it's completely different from other animal’s feces. Therefore, we can distinguish the feces samples of giant pandas from those of other animals.
line 98: randomly
Response:Suggestion accepted, ‘representative’ have been replaced with ‘randomly’(See line 116)
lines 205-206 correct grammar "Here samples used for shotgun..."
Response:Suggestion accepted, we have corrected the poor grammar(See line 221-223).
line 333 enzyme activity DIFFERENCES? between wild pandas
Response:There is a certain individual difference in enzyme activity between wild giant pandas. However, it was no significant difference from the mean.

Reviewer 2 Report
16-17 Symbol for conservation is an unclear term – suggest removal or clarification
Upgraded from “endangered” to “vulnerable” – I presume this is a reference to the IUCN Redlist: if so, it should say that.
17-18 Is this the global population?
22 Differ between
25 “Reduced adaptation when releasing into the wild” – this phrase is unclear and should be amended
26-27 Significantly higher and lower
30 “Huge concern” is probably not appropriate – amend language
31 If they are released into the wild environment
33 Presumably this is the adaptability of individuals
38-40 Include genus and species name; again, specify where it is categorised as endangered
39 “Save the conservation status” is poor English – amend
42 & 45 Formatting error
44 Define gut microbiome
57 Poor English
58 Poor English
71-75 Age of faecal samples? Does this affect results?
80-83 This could have been confirmed by DNA analysis – this method described here seems unreliable. See 240-243
98 Define subadult
355-357 It would be helpful to discuss the natural diet of pandas and their diet in captivity earlier in the paper
Author Response
6-17 Symbol for conservation is an unclear term – suggest removal or clarification
Response:Suggestion accepted, we have revised it(See line 16)
Upgraded from “endangered” to “vulnerable” – I presume this is a reference to the IUCN Redlist: if so, it should say that.
Response:We have added the reference to define this quoted from the IUCN Redlist (See line 18)
17-18 Is this the global population?
Response:Yes, we have defined it(See line 19)
22 Differ between
Response:Suggestion accepted, “in” have been replaced with “between”(See line 23)
25 “Reduced adaptation when releasing into the wild” – this phrase is unclear and should be amended
Response:Suggestion accepted, we have amended this(see line 26-27)
26-27 Significantly higher and lower
Response:Suggestion accepted, “high and low” have been replaced with “higher and lower”(see line 29)
30 “Huge concern” is probably not appropriate – amend language
Response:Suggestion accepted, “Huge concern” have been replaced with “great concern”(see line 32)
31 If they are released into the wild environment
Response:According to the reviewer’s suggestion, we have amended it(see line 33).
33 Presumably this is the adaptability of individuals
Response:Great point! It is hard to avoid the different of the adptability of individuals. We just hope to minimize or eliminate those differences of gut microbiome between captive and wild giant pandas prior to release into the wild.
38-40 Include genus and species name; again, specify where it is categorised as endangered
Response:We have added genus and species name (Ailuropoda melanoleuca)(see line 41); We also cited the IUCN Redlist where it is categorised as endangered(see line 42).
39 “Save the conservation status” is poor English – amend
Response:“save the conservation status of” have been replaced with “protect”(see line 42)
42 & 45 Formatting error
Response:We have amended this error(see line 47-48).
44 Define gut microbiome
Response:Suggestion accepted, we have defined gut microbiome with “gut microbiome of giant pandas”(see line 49)
57 Poor English
Response:We have amended this sentence(see line 65-68).
58 Poor English
Response:We have amended this sentence(see line 65-68).
71-75 Age of faecal samples? Does this affect results?
Response: We provided the information of age of captive giant pandas in Table S1. It is difficult to estimate the age of wild giant pandas by their fecal samples. To avoid the impact of age, we selected captive giant panda of different age stages (including Cub, Sub adult, adult and old) to compare with wild giant pandas. Therefore, age doesn't affect our conclusion.
80-83 This could have been confirmed by DNA analysis – this method described here seems unreliable. See 240-243
Response:Great point! We agree with the reviewer’s point. We redetermined individuals of wild giant pandas by DNA analysis, and a total 27 wild giant pandas were identified according to unique genotypes. We also reanalyze the data of 27 wild giant pandas and 49 captive giant pandas. The results still supported that the diversity and gut microbial communities of the wild and captive giant pandas were significantly different. We have revised those content in methods (line 90-98) and results section(line 258-261, Figure S2 and Table S2).
98 Define subadult
Response:Suggestion accepted, we have defined cub, subadult and adult (see 113-114, Table S1).
355-357 It would be helpful to discuss the natural diet of pandas and their diet in captivity earlier in the paper
Response:Thank you for the reviewer’s excellent suggestions. We have discussed the difference of diet between captive and wild giant pandas before this content(see line 373-377) .
